# Door to Disposition Key Performance Indicator in Three Saudi Arabian Hospitals’ Emergency Departments during COVID-19 Pandemic

**DOI:** 10.3390/healthcare10112193

**Published:** 2022-11-01

**Authors:** Ranya S. Almana, Abeer Alharbi

**Affiliations:** Health Administration Department, Business Administration College, King Saud University, Riyadh 11451, Saudi Arabia

**Keywords:** COVID-19, emergency department, key performance indicator, Door to Disposition, Saudi Arabia

## Abstract

Objective: The COVID-19 pandemic impacted health systems and Emergency Departments (ED) services worldwide. This study attempts to assess the impact of COVID-19 on the performance of the emergency department during COVID-19 in three hospitals in Riyadh city, Saudi Arabia. Methods: Ada’a data was used for this retrospective cohort study. The hospitals included in this study were: a 300-bed maternity and children’s hospital; a 643-bed general hospital; and a 1230-bed tertiary hospital. All patients who visited the ED in the time period from September 2019 to December 2021 were included. The outcome variable was the Door to Disposition (DTD) which estimates the percentage of patients seen within 4 h from Door to Disposition. A two-way ANOVA test was used to examine the differences in the outcome variable by hospital and by the phase of COVID-19. Results: Both hospital and the phase of COVID-19 were significantly different in terms of the percentage of patients seen within four hours in the ED (DTD) (*p*-value < 0.05). On average, the DTD percentages dipped slightly in the early phase of COVID-19 (64.0% vs. 69.8%) and jumped sharply in the later phase (73.6%). Additionally, the average DTD score for the maternity and pediatric hospital (87.6%) was sharply higher than both general and tertiary hospitals (63.2%, and 56.5%, respectively). Conclusion: COVID-19 led to a significant drop in emergency department services performance in the early stage of the pandemic as patients spent more time at the ED. However, for the designated COVID-19 hospital, the ED performance improved as more patients spent less than 4 h at the ED in the early stages of COVID-19. This is a clear indication that careful planning and management of resources for ED services during a pandemic is effective.

## 1. Introduction

The coronavirus disease (COVID-19) intensified worldwide and was declared a pandemic by the World Health Organization (WHO) in early 2020 [1]. The virus was first reported in Wuhan, China, on 13 December 2019. In Saudi Arabia, the first case was confirmed in March 2020 with the number of such cases reaching its peak in June 2020 and fluctuating during July of that year [2]. Then, from August 2020, the number of confirmed cases declined and reached the minimum in February 2021 [2]. From 2 March to 25 July in 2020, there have been 262,772 confirmed cases of COVID-19 with 2672 deaths [3]. Saudi Arabia has had a similar experience with the Middle East respiratory syndrome coronavirus (MERS-CoV) [4], which helped the health system prepare hospitals in the country to face the COVID-19 pandemic. The pre-established presence of patient isolation facilities, personal protective equipment, sterilization tools, and specialized health care workers was crucial to handle the sudden increase in demand for healthcare resources after COVID-19 [4]. On 8 March 2020, the healthcare system in Saudi Arabia enacted a range of mitigation strategies to prevent the spread of the disease. Initial public health strategies adopted included minimizing nonessential hospital visits, especially for children and more vulnerable persons, to decrease the spread of the virus and to ensure that there was enough capacity to handle the surges in COVID-19 cases. Other strategies included the launch of a mass screening program for the early detection and immediate control of the spread of COVID-19. The program focused on screening individuals in highly populated districts through field tests and using a mobile app with a self-assessment tool, which classifies users as low or high risk. The low-risk group was the targeted population and was screened in designated primary care centers [5,6]. The suspected COVID-19 cases with no symptoms were screened at specialized drive-through testing centers [7]. The implementation of specialized fever clinics was another initiative devoted to screening people suspected of having COVID-19 without prior appointments. This service was free of charge and provided diagnostic tests and therapeutic options for people with suspected COVID-19 symptoms, such as fever, shortness of breath, cough and other common clinical characteristics of COVID-19. These specialized fever clinics had an immediate effect on relieving the pressure on emergency departments [8]. Other public health-related mandates consisted of complete and partial curfews, the closure of educational institutions, social distancing, the mandatory use of masks, and the suspension of mass gatherings such as praying in mosques, domestic and international flights, and social and sports events. The COVID-19 pandemic impacted health systems and Emergency Departments (ED) services worldwide [9,10,11,12]. Patients with suspected COVID-19 needed to be separated from others, the staff had to wear protective gear that limited their productivity, and vital parameters had to be re-evaluated with high frequency. All of this increased the workload which led to crowding in the ED [13,14,15,16]. On the other hand, there was a decreased volume of patients seeking health care [17,18] or visiting ED during the COVID-19 pandemic, especially those presenting as lower acuity [19,20,21,22,23].

To assess the complex system of the ED, it is helpful to use the conceptual model introduced by Asplin et al. [24]. The model categorizes the emergency care system into three main components: input, throughput, and output. The input component includes factors that impact the demand for ED services, such as the patient’s need for health care services. The throughput component focuses on the ED internal care processes including triage, physician assessment, diagnostics, and treatment. The output component highlights factors related to the patients’ disposition and the department’s ability to admit them in a timely manner to in-patient care or to safely discharge them. The Ministry of Health (MOH), the largest provider of healthcare services in Saudi Arabia, through the Ada’a (*performance*) program collects Key Performance Indicators (KPIs) such as a means of evaluating the progress of the health system in terms of increases in its capacity, access to healthcare, and reduction in waiting times. The KPIs were developed to be utilized by the hospitals to monitor, evaluate, and improve their performance against established benchmarks. The KPI “Door to Disposition” (DTD) is an output measure that estimates the percentage of patients seen within 4 h from Door to Disposition (i.e., door to doctor to decision to disposition). 

This study attempts to assess the performance of the emergency department during COVID-19 by comparing the “Door to Disposition” rate before and after COVID-19 in three MOH hospitals in Riyadh city, Saudi Arabia. This was conducted with a view of helping policy makers to better understand the impact of the pandemic on the health system so they can prepare contingency plans for any such future pandemics. In addition, the study included three hospitals with different scopes of services which are maternity and children’s hospital, a general hospital, and a tertiary hospital. This inclusion provides an opportunity to assess the performance of EDs in different types of health care settings in order to better understand the situation and prepare suitable plans to face future pandemics.

## 2. Materials and Methods

### 2.1. Data Source

We used Ada’a data collected from three Saudi hospitals for this retrospective cohort study. The hospitals included in this study were: a 300-bed maternity and children’s hospital (MCH) that offers two emergency services; maternity and pediatrics; a 643-bed general hospital (GEN) that offers general emergency services; and a 1230-bed tertiary hospital (MDC) that offers three emergency services—adult, maternity, and pediatrics. Data were collected on a monthly basis. We used data from all patients who visited the ED in the time period from September 2019 to December 2021. 

### 2.2. Outcome Variable

The outcome variable used for this analysis was the Door to Disposition (DTD), which is calculated by dividing the number of patients served within four hours by the total number of patients visiting the ED in seven days multiplied by 100. Disposed patients were either admitted, discharged to go home, or transferred to another facility. Table 1 provides details on the sample size required to be collected, the inclusion and exclusion criteria, and the benchmark classification and target.

### 2.3. Statistical Analysis

The data were analyzed using the Statistical Package for Social Sciences (SPSS) program, version 25. A two-way ANOVA test was used to examine the differences in the outcome variable by hospital and by the phase of COVID-19 which was divided into three phases: Pre-COVID-19 (September 2019–February 2020) coded as 0; early-COVID-19 (March 2020–July 2020) coded as 1; and later-COVID-19 (August 2020–December 2021) as 2. A *p*-value of less than 0.05 was considered to be statistically significant. Effects plots with fitted means and confidence intervals (CI) were used to explain the difference in the outcome variable by hospital and by the phase of COVID-19. 

### 2.4. Ethical Considerations

Institutional Review Bord (IRB) approval for the study was obtained from King Fahad Medical City (KFMC), reference number 22-347E. 

## 3. Results

The data were collected over 28 months beginning in September of 2019. The percentage of patients seen within four hours in the ED (DTD) was plotted by hospital across the 28-month period (Figure 1). To examine the differences in the outcome variable by hospital and by the phase of COVID-19, a two-way ANOVA test was used. The results showed that both hospital and the phase of COVID-19 were significantly different in terms of the percentage of patients seen within four hours in the ED (DTD) (*p*-value < 0.05) (Table 2). The Effects plots show how they were different [Figure 2]. On average, the DTD percentages dipped slightly in the early phase of COVID-19 (64.0% vs. 69.8%) and jumped sharply in the later phase (73.6%). Additionally, the average scores for the maternity and children’s hospital (MCH) (87.6%) were sharply higher than both the general hospital (GEN) and the tertiary hospital (MDC) (63.2%, and 56.5%, respectively). Looking at the interaction plots, the percentage for DTD dropped during the early phase of COVID-19 for both the maternity and children’s hospital (MCH) (75.2% vs. 95.5%) and tertiary hospital (MDC) (46.5% vs. 56.5%), while the general hospital (GEN) percentages rose distinctly (70.4% vs. 57.3%). Then in the later phase, both MDC and MCH rose to the pre-COVID-19 level (66.7%, and 92.2, respectively) while GEN dropped slightly (62.1%).

## 4. Discussion

Public health-related mandates recommended social distancing and minimizing unnecessary hospital visits to prevent the spread of the disease. Emergency departments’ mitigation strategies included setting up and equipping triage, establishing a separate waiting area for suspected COVID-19 patients, and making sure health workers adhered to safety precautions which included hand hygiene and wearing masks. All of these new mitigation strategies were likely to have an effect on the waiting times in the emergency departments. This study aimed at assessing the effect of COVID-19 on the performance of the ED in terms of the percentage of patients seen within 4 h, i.e., Door to Disposition (DTD). The study also explored whether the type of hospital affected the ED performance. 

The study findings revealed that there was a variation in the ED performance before and after COVID-19. Overall, during the early stage of COVID-19, the percentage of patients seen within 4 h in the ED decreased by 6 percentage points. The prolonged waiting time at the ED was probably caused by the extra precautionary steps implemented in response to the spread of the disease and from patients having to wait for examination results [13,14,15,16]. The changes implemented at the Saudi hospitals in this study included patients presented in the ED with suspected COVID-19 having to be separated from other patients; infected patients being isolated in hospitals or hotels; the staff having to wear protective gear that slowed down their productivity; and the need for the vital parameters to be re-assessed more than once. All of these activities led to an increased workload and resulted in long waiting times in the ED. Consequently, patients with high acuity illness or injury may have had to wait far too long to be adequately treated. Previous studies had found that patients with lower acuity were avoiding visiting ED during the COVID-19 pandemic in Saudi Arabia [19,21] and other health systems [18,19,21,23,24]. In our study population, the number of ED visits declined after COVID-19 [Appendix A]. Although, the number of ED visits declined after COVID-19, the ED performance was affected negatively. These findings are consistent with previous studies that discovered that the patient flow was negatively affected by higher acuity despite a decreased number of ED visits in Saudi Arabia [21] and USA [25]. In the later stage of COVID-19; that is from the months of August 2020 to December of 2021, the percentage of patients seen in ED within 4 h increased sharply by 9 percentage points. This improvement most likely came with the post-lockdown abatement of COVID-19. 

The study findings also revealed that the ED performance seemed to vary significantly across hospitals. This was not surprising considering the dissimilarity in the scope of services provided in each of them, the population served, and the complexity of the medical condition. In addition, the general hospital (GEN) was assigned as a COVID-19 hospital while the other two were not. The 300-bed maternity and children’s hospital (MCH) seemed to have the best ED performance. Its overall DTD percentage was scored as “acceptable” according to the KPI benchmark, with more than 87 percent of patients being seen within four hours at the ED across the 28-month study period. MCH offers only two emergency services, namely, maternity and pediatrics. The low acuity of patients presenting at the ED might be the reason why more patients were seen within 4 h compared to the other two hospitals that accommodated more complex patient conditions. The tertiary hospital (MDC) showed the lowest ED performance among the three with its overall DTD rated as “unacceptable”, according to the KPI benchmark, with less than 60% of patients being seen within four hours across the 28-month study period. MDC is a large 1230-bed tertiary hospital that offers emergency services for adults, maternity, and pediatrics. The complexity of cases presented at the ED might explain the longer waiting times there [26]. Similarly, the 643-bed general hospital (GEN), which offers general emergency services, fell into the category of “need improvement” for its overall DTD performance. This was based on the KPI benchmark that showed less than 65 percent of patients being seen within four hours at the ED across the 28-month study period. Both MDC and GEN provide ED services for more complex cases than MCH. Crowdedness and being busy with suddenly deteriorated patients were found to be the top reasons for delay in the ED [21]

Looking at the effect of COVID-19 on each hospital, we observed that during the early stage of COVID-19, both the maternity and children’s hospital (MCH) and the tertiary hospital (MDC) ED performance deteriorated. The DTD percentage dropped sharply for MCH going from “word class” at pre-COVID-19 to barely “acceptable” in the early COVID-19 stage. In addition, MDC performance dropped by 10 percentage points in early COVID-19. These drops in DTD rates were to be expected, as health systems, and particularly EDs, were affected by the pandemic with the surges of COVID-19 cases and the newly implemented guidelines and mitigation precautions implemented by the ED. All of these occurrences resulted in prolonged processes and increased waiting times. In the later stage of COVID-19, however, both MCH and MDC improved their DTD rates with the post-lockdown abatement of COVID-19. Nevertheless, and contrary to both MCH and MDC, the general hospital (GEN) performance actually improved in the early stages of COVID-19 as the percentage of patients seen within 4 h increased. This was probably because GEN was closed for five months in 2020 and seven in 2021 and was turned into a designated COVID-19 hospital focusing on critical care. As a consequence, there was careful planning and management of resources in preparation for the surge of potential patients with the highest acuity. It was, therefore, possible to avoid ED crowding, even during extreme conditions, by leveraging previously known input, throughput and output factors, including a change in working methods with higher competence, less diagnostics, and increased focus on rapid clinical admission decisions [14,27]. 

## 5. Limitations

This study was conducted in the ED of a specific geographical area of Saudi Arabia which may affect the generalizability of the study. In addition, due to the study’s retrospective nature, limited information was collected on some areas. We used a database of the quality metrics and not medical charts; therefore, the lack of individual patient characteristics prevented us from performing a further analysis as to the possible reasons for the prolongation of the DTD process in terms of diagnosis, acuity level, and demographics. Future studies that would take these factors in consideration are required.

## 6. Conclusions

The current study indicates that COVID-19 led to a significant drop in emergency department services performance in the early stage of the pandemic, as patients spent more time at the ED when compared to before the pandemic. A critical consequence of this happening was that patients with high acuity might have had to wait too long to be treated adequately. On the other hand, for the designated COVID-19 hospital, the ED performance improved as more patients spent less than 4 h at the ED post-COVID-19. These findings have provided a better understanding of the impact of the pandemic on the health system and should allow policy makers to prepare contingency plans for any such future pandemics. 

## Figures and Tables

**Figure 1 healthcare-10-02193-f001:**
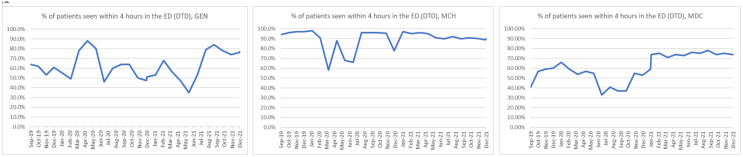
Percentage of patient seen within four hours in the emergency department (DTD) over 28 months in three hospitals (GEN, MCH, MDC).

**Figure 2 healthcare-10-02193-f002:**
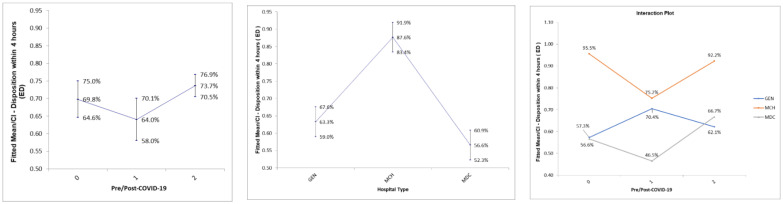
Effects plots for the percentage of patient seen within four hours in the emergency department (DTD) by hospital (GEN, MCH, MDC) and COVID-19 phase (pre (0), early (1), late (2)).

**Table 1 healthcare-10-02193-t001:** Data collection/Measurement in Emergency Department.

KPI: Door to Disposition (Percentage of Patient Seen within Four Hours)	
Sample size	All patient volume
Numerator	Number of patient Door to Disposition < 4 h.
Denominator	Total number of patients
Calculation	[Numerator/denominator] × 100
Exclusion	DAMA *, LAMA ** & Morgue
Disposition	patient either Admission or Discharge
Patient admission	ICU and WARD
Patient discharge	DAMA, LAMA, Home, Transferred to another facility and deceased.
World class	more than 95%
Acceptable	75–95%
Need Improvement	60–75%
Unacceptable	less than 60%

* DAMA: Discharge Against Medical Advice; ** LAMA: Leave Against Medical Advice.

**Table 2 healthcare-10-02193-t002:** Analysis of Variance (ANOVA).

Source	DF	SS	MS	F	*p*
Pre/Post-COVID-19	2	0.106306131	0.053153066	4.121	0.0200
Hosp. Type	2	1.109494876	0.554747	43.007	<0.001
Interaction	4	0.238680	0.05967011	4.626	0.0022
Error	75	0.967428	0.012899034		
Total	83	2.764	0.033304285		

## Data Availability

Available upon request.

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
