# Peer review of "Door to Disposition Key Performance Indicator in Three Saudi Arabian Hospitals’ Emergency Departments during COVID-19 Pandemic"

_healthcare, 2022, doi:10.3390/healthcare10112193_

Round 1

Reviewer 1 Report

interesting paper. Grammar and syntaxis are unremarkable. I have only a perplexity about comparing three hospitals with different peculiarities, purposes, and number of beds one of which was closed for a very long period in both 2020-2021 and totally converted.

Author Response

Thank you so much for your comment. The three hospitals included in the study were hospitals that operated under the umbrella of the Ministry of Health (MOH) that participated in the Ada’a program that measures quality performance. The first one is MCH hospital, a 300-bed maternity and children’s hospital that offers two emergency services; maternity and pediatrics. The second hospital is GEN hospital, a 643-bed general hospital that offers general emergency services. GEN hospital was designated as a COVID-hospital. This was done with the intention that careful planning and management of resources in preparation for the surge of potential patients with the highest acuity during pandemic is effective. GEN was selected as a COVID-hospital and was closed for longer time because it offers general services rather than specialized ones, which are available in other MOH hospitals that would fulfill these needs. The third hospital included in this study is MDC hospital, a 1230-bed tertiary hospital that offers three emergency services - adult, maternity, and pediatrics.

Author Response

Thank you so much for your comments. Below are the responses for each comment:

  • "The COVID-19 pandemic impacted health care systems and Emergency Department services worldwide." It is OK generally. But it would be interesting to know whether the authors agree that the "health care system" and "health system" are different. It is possible to find that the distinction between them lies in health determinants: the health system encompasses wider determinants of health, and the health care system is limited to personal care.

Response:

Yes, we agree that the "health care system" and "health system" are different. Therefore, we decided to change it to (Health systems). These changes although might be minor but would help create a "common language" in public health as you suggested.

  • Probably Brent A. Asplin's propositions are relevant for the aims of this study. However, it would be exciting to consider the model of the Kingdom of Saudi Arabia and the country's specialite de la Maison. If existed.

Response:

We decided to use Asplin's model to explain the ED components because we felt it was a sound model that apply to the ED in Saudi hospitals, and would describe our output measure (Door to Disposition) in the context of input, throughput, and output.

  • Firstly, I have nothing against Cucinotta et al. paper on the WHO declaration, but I am pretty sure it would be much better to consider the original WHO publications.

Response:

We cited the original WHO publication.

  • Authors move to the data on the COVID19 situation in Saudi Arabia. It is OK. But the next step is the information on the international cases. Based on papers dealing with the situation in Saudi Arabia, Greece, France /Eastern, and Italy. What about Portugal, the Czech Republic and others? What was the key to choosing these countries? The case of the Kingdom of Saudi Arabia is self-explanatory. But why not Greece and Slovakia, for example?

Response:

For the literature review, we used key words for the search that included: emergency department, COVID-19, impact, waiting time, door to disposition, etc. and tried to cite the publications that had similar aims and provided results and conclusions that is related to our study. We included all papers we found that we believed fit our aims which was not specific to a geographic region but came from different parts of the word.

  • But let's start with the title. The story is about the impact of COVID-19 on the emergency department's key performance indicator in three hospitals in Saudi Arabia. But, do not you think that viruses impact the pandemic situation., so to speak, rather than the indicators? Indicators indicate the status that is a result of the virus outbreak. Hence, I am pretty sure that the title should be adjusted a bit, for example: "Fluctuation of Key Performance Indicator in Saudi Arabian Hospital ED in COVID- 19 pandemic time." or "Door to Disposition Key Performance Indicator in Saudi Arabian Hospital Emergency Department in COVID-19 Pandemic."

Response:

The authors agree with this comment. Therefore, the title is adjusted to “Door to Disposition Key Performance Indicator in Three Saudi Arabian Hospitals’ Emergency Departments during COVID-19 Pandemic”

  • As the structure of the introduction is concerned, I would like to offer the following layout: 1) A few words on the COVID-19 epidemic globally and specifically in the Kingdom of Saudi Arabia. 2) A few sentences on the country's health system organization and performance with information on mitigation procedures and proper ED issues, e.g. data on organization and indicators. 3) The aim of the study. Maybe it can help policymakers better understand the pandemic's impact on social life. But the information on choosing hospitals is much more critical here in the paper. It is one of the essential data in this study. The statement that they have a different scope of services is far too little.

Response:

Added accordingly in the main text (lines 34-44 and 49-61 and 91-95)

  • The part named "Discussion: The place for some general information is in the introduction. It is not necessary to repeat them. For example, information that health care systems have been affected.

Response:

We agree with you. Therefore, the repetitive sentence has been omitted accordingly.

  • When the authors refer an issue to other publications, it would be better to inform readers about sources in detail. Let's consider such an example. There is a description of anti-epidemic measures in The Kingdom of Saudi Arabia's hospitals. And after that, the authors give the readers information about "previous" studies. But it is a difference between "previous" regarding local hospitals and other countries. The same situation in the country means less than observed in a specific region or globally. There are a few such cases in this part of the paper.

Response:

The paragraphs have been rephrased in order to include studies from the Saudi context then introduce other studies from international context that would support our findings. (these changes are highlighted in the main text lines 150-152 and 155-157)

  • But it is challenging to agree with the authors that this study indicates that careful planning and management of resources for ED services during a pandemic is effective. Probably is. But again, it is general wisdom. Not as the result of this study. I would be for deleting such a conclusion.

Response:

We will delete this conclusion. We reached this conclusion by observing the improved results from that one hospital that was converted to a COVIS-hospital. However, because this was not explained clearly in the aims or methods, we decided to delete this conclusion.

Reviewer 3 Report

The manuscript “The Impact of COVID-19 on the Emergency Department Key Performance Indicator in Three Hospitals in Saudi Arabia” presents a retrospective cohort study evaluating the performance indicator in three emergency department in Saudi Arabia.

In the manuscript, the question is original and well defined and the results provide an advance in current knowledge; the results are interpreted appropriately; all conclusions are justified and supported by the results; the article is written in an appropriate way; the data and analyses are presented appropriately; the study is correctly designed and technically sound; the analyses are performed with the highest technical standards; the methods, tools, software, and reagents are described with sufficient details to allow another researcher to reproduce the results; the conclusions are interesting for the readership of the Journal and the paper presumably will attract a wide readership; there is an overall benefit to publishing this work; the English language is appropriate and understandable.

The only comments that I would like to suggest to the authors are the following:

1.     Specify the acronym ADA'A

2.     Expand and deepen the introduction and discussion with the following topics:

·      Increased mortality among doctors (https://doi.org/10.3390/healthcare10091684; https://doi.org/10.3390/healthcare10071187)

·      Healthcare Avoidance during COVID-19 (https://doi.org/10.3390/healthcare10071261; https://doi.org/10.3390/healthcare10081360)

·      Particular healthcare issues affected by the COVID-19 pandemic (https://doi.org/10.3390/healthcare10061018)

·      Reduction in Hospital Admissions (https://doi.org/10.3390/healthcare10050871; https://doi.org/10.3390/healthcare10040639; https://doi.org/10.3390/healthcare10081376)

·      Physician Engagement (https://doi.org/10.3390/healthcare10081394)

·      The need to routinely conduct clinical audits to identify clinical challenges and make recommendations for health promotion (https://doi.org/10.3390/healthcare10071338)

·      Medical and Health Resources Allocation in the Context of COVID-19 (https://doi.org/10.3390/healthcare10071319)

For the reasons listed above, my final recommendation is to accept after minor revisions the manuscript.

Best regards

Author Response

  • Specify the acronym ADA'A

Response:

Thank you so much for your comments. ADA'A is not an acronym, it is an Arabic word that means performance which is the name of the program. to avoid the confusion, we will re-write it to Ada’a (only capitalize the first letter), and add in parenthesis the meaning which is (performance)

  • Expand and deepen the introduction and discussion with the following topics:

  Increased mortality among doctors (https://doi.org/10.3390/healthcare10091684; https://doi.org/10.3390/healthcare10071187)

      Healthcare Avoidance during COVID-19 (https://doi.org/10.3390/healthcare10071261; https://doi.org/10.3390/healthcare10081360)

     Particular healthcare issues affected by the COVID-19 pandemic (https://doi.org/10.3390/healthcare10061018)

     Reduction in Hospital Admissions (https://doi.org/10.3390/healthcare10050871; https://doi.org/10.3390/healthcare10040639; https://doi.org/10.3390/healthcare10081376)

      Physician Engagement (https://doi.org/10.3390/healthcare10081394)

     The need to routinely conduct clinical audits to identify clinical challenges and make recommendations for health promotion (https://doi.org/10.3390/healthcare10071338)

   Medical and Health Resources Allocation in the Context of COVID-19 (https://doi.org/10.3390/healthcare10071319)

Response:

These references were added accordingly in the main text (lines 69-71 and lines 65-65 and lines 173-175 and lines 222-225)